# Development of the awareness, skills, knowledge: General (ASK-G) scale for measuring cultural competence in the general population

**Melanie M. Domenech Rodríguez**[1]*, **Alexandra K. Reveles**[2], **Kaylee Litson**[3], **Christina A. Patterson**[4], **Alejandro L. Vázquez**[1]

**1** Department of Psychology, Utah State University, Logan, Utah, United States of America, **2** LifeStance Health, Corvallis, Oregon, United States of America, **3** Department of Instructional Technology and Learning Sciences, Utah State University, Logan, Utah, United States of America, **4** Counseling and Psychological Services, University of Nevada, Las Vegas, Nevada, United States of America

* melanie.domenech@usu.edu

**Data Availability Statement:** The data and accompanying files are available on OSF at this

## Abstract

Measuring cultural competence has been difficult for conceptual and practical reasons. Yet, professional guidelines and stated values call for training to improve cultural competence. To develop a strong evidence-base for training and improving cultural competence, professionals need reliable and valid measures to capture meaningful changes in cultural competence training. We developed a measure for cultural competence that could be used in a general population to measure changes in awareness, knowledge, and skills in interacting with culturally diverse others. We built an 81-item scale with items conceptually categorized into awareness, knowledge, and skills and was presented to an expert panel for feedback. For evaluation, a national panel of 204 adults responded to the new scale and other measures associated with cultural competence. Factor analysis revealed four factors with strong reliabilities: Awareness of Self, Awareness of Others, Proactive Skills Development, and Knowledge ($as$ = .87 - .92). The final overall scale, Awareness, Knowledge, Skills—General (ASK-G) had 37 items and strong reliability ($a$ = .94). The ASK-G was then compared to validated scales to provide evidence of concurrent, convergent, and divergent validity. Strong evidence emerged for these. The ASK-G is a promising tool to measure cultural competence in a general population.

## Introduction

Following the devastating deaths of George Floyd, Breonna Taylor, Ahmaud Arbery, and other Black, Indigenous, and other People of Color (BIPOC) and the subsequent months-long protests that occurred in the summer of 2020, many find themselves wondering how we came to be at this point in U.S. history [1]. Tensions have continued to flare with significant political backlash against teaching Critical Race Theory in schools and denialism surrounding the

link: https://osf.io/quxa3/ and the assigned doi is: https://doi.org/10.17605/OSF.IO/QUXA3.

**Funding:** The author(s) received no specific funding for this work.

**Competing interests:** The authors have declared that no competing interests exist.

existence of systemic racism [2]. At the same time, the requests for equity, diversity, and inclusion trainings have also dramatically increased. Psychologists possess unique expertise to address this need, yet cultural competence assessments are tailored for counselors, graduate students in psychology and education programs, and healthcare professionals. While cultural competency measures for the general population exist, several limitations hinder their utility in assessing training and research outcomes (i.e., narrow focus on specific components, poor predictive validity, and generalizability of factor structure) [3]. The current study sought to address this need through the rigorous development of a scale to measure self-reported cultural competence with an emphasis on race and ethnicity in a general population. The Awareness, Skills, Knowledge: General (ASK-G) measure developed in this study drew from Sue's operational definition of cultural competence and, in particular, sought to measure awareness, knowledge, and skills in a general population [4].

## Cultural competency

Public and private universities, government agencies, and professional guilds espouse values to advance diversity [5, 6]. However, the very use of the word diversity has been contested as political and loaded [7]. Although there is some evidence of effectiveness for interventions to improve cultural competence [8], emerging evidence is still in early stages and quite limited [9, 10]. Accruing evidence of effectiveness is partly hindered by the difficulty in measuring cultural competence [11, 12] and who provides the assessment of competence (e.g., self or client) [11]. This is partially due to differences in how terms are defined [12, 13], and how to measure these debated concepts.

Definitions of cultural competence vary across and within disciplines. A recent review listed 35 definitions of cultural competence in the helping professions alone [14]. For instance, the Association for American Medical Colleges defined cultural and linguistic competencies as sets of "congruent behaviors, knowledge, attitudes, and policies that come together in a system, organization, or among professionals that enables effective work in cross-cultural situations" (p. 1) [15]. The U.S. Department of Health and Human Services defines cultural competence as "services [that] are respectful of and responsive to the health beliefs, practices, and needs of diverse patients" and has provided standards to promote health equity, enhance the quality of healthcare, and help eliminate disparities in healthcare [6, 16]. These standards highlight the importance of individual changes in service provisions (e.g., offering language assistance, awareness of cultural influences) and more broadly, changes in policy [6].

Within psychology, cultural competence has been defined as "the belief that people should not only appreciate and recognize cultural groups but also be able to effectively work with them" (p. 440) [4]. This definition has been widely used by mental health practitioners and scholars interested in diversity issues, and encompasses three dimensions of cultural competence: awareness, knowledge, and skills [4]. These dimensions comprise the so-called tripartite model of cultural competence. *Awareness* refers to the person's recognition of belonging to a cultural group and allows for self-examination of values, beliefs, and practices in a manner that enhances humility and facilitates empathy. Awareness also includes understanding that there are others that are culturally different than oneself [17]. The *knowledge* dimension refers to acquiring and retaining information specific to cultural groups. Knowledge could be language (e.g., words, phrases, proficiency), specific traditions (e.g., practices around childbirth), or rules for interpersonal exchanges (e.g., whether or not to shake hands). Finally, *skills* refer to communicative or behavioral repertoires that result in successful exchanges between culturally different people.

Outside of the healthcare disciplines, scholarship in business describes techniques and etiquette promoting diversity and inclusion–rather than defining cultural competence–in communication or business relationships. Through this lens, diversity is articulated as variety in domains such as the heritage, background, and tendencies of people in the workplace that includes age, race/ethnicity, culture/nationality, sexual orientation, religion, veteran status, ability status, neurodiversity, education, socioeconomic status, worldview, and lifestyle [18]. This description of cultural competence is similar to that of healthcare with their shared emphasis on valuing differing perspectives, as well as the focus on skills, however the definitions remain distinct to each field.

Because of the diversity of definitions across and within disciplines, it has been difficult to conceptualize or measure cultural competence. Various concepts that fall under the awareness, knowledge, or skills domain of cultural competence are used as proxies. For example, when examining cultural competence related to race and ethnicity, *colorblindness* (denial of racism and denial of judgement of others based on race) has been conceptualized as a facet of awareness and *ethnocultural empathy* (empathy for another person from a different race/ethnicity) has been conceptualized as a facet of skills [19–21]. Other constructs have been conceptualized as influencing the ability to develop awareness, knowledge, or skills. Social dominance orientation has been inversely related to multicultural knowledge [22]. Personality factors have also been linked to various facets of prejudice or openness to diversity, such as openness to experience being related to lower prejudice [23] and appreciation of cultural diversity [24].

While multi-dimensional measures of cultural competency exist, they are often designed for professional populations and/or focus on specific aspects of cultural competency [3]. Cultural competency measures commonly include factors pertaining to professionals' awareness, knowledge, and skills (e.g., Multicultural Counseling Inventory, Inventory for Assessing Process of Cultural Competence among Health Professionals; Multicultural Awareness, Knowledge, and Skills Survey) [25–27]. Other measures for professionals include some of these components (e.g., Cultural Self-efficacy Scale, factors representing knowledge and skills; Multicultural Counseling knowledge and Awareness Scale, factors resenting knowledge and awareness) or focus on other factors broadly associated with cultural competency (e.g., Cultural Competence Self-Assessment Questionnaire, factors representing knowledge of community, personal involvement, service delivery, proactive, community outreach, etc.) [28, 29]. Two measures exist that are intended for general populations but focus specifically on attitudes relating to racial/ethnic groups and women's equity issues (i.e., Quick Discrimination Index) or on proxies for cultural competency and have questionable psychometric properties and utility (i.e., Cross-Cultural Adaptability Inventory; factors representing emotional resilience, flexibility/openness, perceptual acuity, personal autonomy) [30, 31]. These studies suggest that cultural competency measures are often intended for professionals, may not assess components of cultural competency as conceptualized in the psychological literature (i.e., awareness, knowledge, skills), and/or may have questionable psychometric properties and utility.

The literature suggests a lack of measures for assessing multiple dimensions of cultural competency within the general population as conceptualized in the psychological literature (i.e., awareness, knowledge, skills) [4]. This may contribute to researchers relying on proxy measures of cultural competency in the general population such as colorblindness [32], empathy [33], and social dominance orientation [34]. Each of these scales only explores specific facets of cultural competence, and there are limitations to this practice. Examinations of cultural competence typically only measure one aspect of cultural competence [21] and measures are developed for specific audiences. A review of popular measures of cultural competence in the health professions revealed all but two measures were developed for professionals that either had a narrow focus on specific components or had questionable utility [3]. The current study

sought to address this measurement gap through the development of the ASK-G scale to measure broad cultural competence related to race and ethnicity for a general population. Cultural competency is conceptualized and defined differently across fields. As we sought to develop the current scale to support psychologists in evaluating cultural competency in training and research we drew from the Sue's cultural competency model to develop ASK-G items, factors loadings (awareness, knowledge, skills), and while seeking expert feedback to form the final scale [4]. If the theorized three-factor loading fit the data poorly using a confirmatory approach, an exploratory analyses would be conducted to identify factors within the scale. To improve the generalizability of our findings, we sought to administer the survey to participants who would be recruited to match racial/ethnic representation in U.S. census to examine the factor structure of the ASK-G. Lastly, we sought to establish the concurrent, convergent, and divergent validity of the ASK-G.

## Method

### Participants and study procedures

For the first phase of the study, we sought two waves of expert feedback, we solicited 24 experts and secured 13. The authors selected professionals known for their contributions to scholarship in cultural competence. For those that provided demographic information, experts were between 32 and 60 years of age and identified as women ($n = 4$) and men ($n = 8$) of varied ethnic backgrounds (1 African American, 2 American Indian, 6 Latinx, 1 Asian American, 1 European American, 1 "other"). Once feedback was incorporated into the scale, a second wave of experts was approached for their feedback on the revised scale. Of the 30 experts contacted, 10 were secured. For those who provided demographic information, ages ranged from 39 to 71, and experts reported belonging to varied ethnic groups (1 African American, 1 Latinx, 2 White American, and 1 "other"). Five reported being women and four reported being men. After receiving expert feedback, the final scale was reviewed by the team of researchers. The research team was comprised of one faculty member (a Latinx cisgender woman) and five graduate students with varied ethnic and gender identities, sexual orientations, and socioeconomic backgrounds.

For the second phase of the study, we utilized Qualtrics panels to test the questionnaire. To determine an adequate sample size, we followed guidelines by Marsh and colleagues [35]. They suggested there is a payoff between the number of items used per factor and the required sample size. We expected to find three factors (Sue's cultural competency model; awareness, knowledge, and skills) with six to 12 items per factor and thus determined that $N = 200$ was a sufficient sample size. Once institutional approval was secured from the [masked for review], a survey was uploaded to Qualtrics. Qualtrics was contracted to secure a panel of 200 adults, 18 years of age or older. Participants were to be equally split by gender and include racial/ethnic diversity similar to that of the general population, based on Census statistics. Participants were 204 adults, varying in age between 19 and 78 years ($M = 41.63$, $SD = 16.08$). Our sample approximated the general population as reflected in U.S. Census data. See Table 1 for full demographic characteristics.

### Instrument development

All items in the ASK-G were designed to address awareness, knowledge, or skills. To determine scale items for the general population, a two-step process occurred.

**Step 1.** The research team developed an initial scale with 81 items intended to assess cultural competence, with 25 items assessing awareness, 29 assessing knowledge, and 27 assessing skills. Once the measure was drafted, IRB approval was secured for this research. Experts were

**Table 1. Sample demographic characteristics (*N* = 204).**

| Race/Ethnicity | *n* | % |
|---|---|---|
| White American | 113 | 55.39 |
| Latinx | 31 | 15.20 |
| Mixed ethnic | 24 | 11.76 |
| Black American | 23 | 11.27 |
| Asian American | 9 | 4.41 |
| American Indian | 2 | 0.98 |
| Other | 2 | 0.98 |
| Prefer not to answer | 2 | 0.98 |
| **Gender identity** | | |
| Man, male, or masculine | 93 | 45.59 |
| Transgender man, male, or masculine | 1 | 0.49 |
| Woman, female, or feminine | 102 | 50.00 |
| Gender nonconforming, genderqueer, or gender questioning | 1 | 0.49 |
| Intersex, disorders of sex development, two-spirit, or other related terms | 2 | 0.98 |
| Other, please specify: | 4 | 1.96 |
| Prefer not to answer | 1 | 0.49 |
| **Sexual orientation** | | |
| Heterosexual or straight | 175 | 85.78 |
| Gay or lesbian | 5 | 2.45 |
| Bisexual | 16 | 7.84 |
| Pansexual | 2 | 0.98 |
| Demisexual | 1 | 0.49 |
| Questioning | 1 | 0.49 |
| Asexual | 1 | 0.49 |
| I prefer not to answer. | 3 | 1.47 |
| Social class | | |
| Poor | 31 | 15.20 |
| Working class | 81 | 39.71 |
| Middle class | 83 | 40.69 |
| Affluent | 9 | 4.41 |
| **Level of education** | | |
| Some high school | 5 | 2.45 |
| High school diploma or equivalent | 56 | 27.45 |
| **Vocational training** | 9 | 4.41 |
| Some college | 40 | 19.61 |
| Associate's degree (e.g., AA, AE, AFA, AS, ASN) | 25 | 12.25 |
| Bachelor's degree (e.g., BA, BBA BFA, BS) | 49 | 24.02 |
| Some post undergraduate work | 2 | 0.98 |
| Master's degree (e.g., MA, MBA, MFA, MS, MSW) | 12 | 5.88 |
| Specialist degree (e.g., EdS) | 2 | 0.98 |
| Applied or professional doctorate degree (e.g., MD, DDC, DDS, JD, PharmD) | 1 | 0.49 |
| Doctorate degree (e.g., EdD, PhD) | 3 | 1.47 |
| Other: | 1 | 0.49 |
| **Geographical location** | | |
| Midwest | 40 | 19.61 |
| Northeast | 46 | 22.55 |
| South | 76 | 37.25 |

(*Continued*)

**Table 1.** (Continued)

| Race/Ethnicity | *n* | % |
|---|---|---|
| West | 41 | 20.10 |
| Puerto Rico or other U.S. territories | 1 | 0.49 |
| **Religion** | | |
| Agnostic | 11 | 5.39 |
| Animist | 1 | 0.49 |
| Atheist | 9 | 4.41 |
| Buddhist | 1 | 0.49 |
| Christian (e.g., Catholic, Lutheran, Methodist, Mormon, Presbyterian, Protestant) | 124 | 60.78 |
| Hindu | 1 | 0.49 |
| Humanist | 1 | 0.49 |
| Jewish | 5 | 2.45 |
| Muslim | 2 | 0.98 |
| Polytheist | 1 | 0.49 |
| Spiritual but not religious | 11 | 5.39 |
| Unitarian Universalist | 2 | 0.98 |
| Wiccan | 4 | 1.96 |
| Other, please specify: | 7 | 3.43 |
| Prefer not to answer | 24 | 11.76 |

contacted via email. The email included a link to a letter of information in Qualtrics. Once they agreed to participate, experts rated the 81 items on whether the items measured cultural competence (*strongly agree*, *agree*, *disagree*, *strongly disagree*). If a panelist indicated that they agreed or strongly agreed that an item assessed cultural competence, they were asked what domain of multicultural competence the item assessed (*awareness*, *knowledge*, *skills*). Experts were not provided with definitions for awareness, knowledge, or skills as they were expected to be intimately familiar with the tripartite model of cultural competence.

**Step 2.** After the expert panel reviewed each item, the research team met to revise the scale. Items were retained where 80% of the experts agreed or strongly agreed that the item measured cultural competence. At the revision, the research team eliminated 23 items. Eliminated items are listed in Table 2. The team then added 25 items that addressed skills. After a secondary expert panel review and team review, the scale contained 84 items. Responses were on a 6-point Likert-type scale: (1) *strongly agree*, (2) *disagree*, (3) *slightly disagree*, (4) *slightly agree*, (5) *agree*, (6) *strongly agree*. The prompt used in the Qualtrics survey was: *Rate how much you agree or disagree with the statements below using the following scale.*

**Expert panel commentary.** Some expert reviewers (*n* = 11) provided either open-ended commentaries on the form, sent emails, or provided feedback on the phone. There was notable consensus about ASK-G food, language, and travel questions. Experts noted that these questions could be answered affirmatively by those truly engaged in cultural competence but also by those wishing to claim it through engagement in superficial activities or even engaged in cultural appropriation and/or exploitation. One reviewer eloquently noted:

A lot of fully colonial people have interest in international food and culture. I could think of tons of [people] who travel abroad, having five star experiences, never engaging with the culture on anything more than an entirely superficial level, who would score highly on these items (and should certainly not be judged culturally responsive).

The expert further cautioned:

**Table 2. Items eliminated by first expert panel.**

| Item | Percent agreement |
|---|---|
| 1. I have a desire to travel to unfamiliar places to learn about new cultures. | 40% |
| 2. I know how to speak another language. | 55% |
| 3. I am familiar with some major words and phrases from a language other than my own. | 60% |
| 4. I am familiar with foods from a cultural group other than my own. | 44% |
| 5. I am familiar with aspects of popular culture from a culture other than my own. | 66% |
| 6. I watch television shows or movies from cultures other than my own. | 55% |
| 7. I listen to music from cultures other than my own. | 44% |
| 8. I know how to prepare food dishes from cultures other than my own. | 44% |
| 9. I can name five world leaders outside of the United States. | 66% |
| 10. I can summarize current events from across the globe on a weekly basis. | 44% |
| 11. I can communicate with someone who doesn't speak a language that I speak. | 66% |
| 12. I can communicate with someone that speaks with a strong foreign accent for more than 5 minutes. | 50% |
| 13. There is a little bit of truth to most stereotypes applied to specific cultural groups. | 55% |
| 14. I can try new, unusual foods (e.g., pig's eyes, cow tongue) even if I am a little disgusted by them. | 44% |
| 15. If someone plays unfamiliar music, I ask for it to be turned off or changed. | 33% |
| 16. I have taken a multicultural or diversity class. | 66% |
| 17. I regularly eat cuisine that is from a different culture than my own (e.g., at a restaurant, friend's home, community event). | 50% |
| 18. I like trying new ethnic foods. | 44% |
| 19. I am able to watch foreign language films and understand the storyline. | 33% |
| 20. I can watch foreign language films and enjoy them. | 40% |
| 21. It is ok for people to adopt identities from cultural groups other than their own (e.g., a White American saying she's "a little bit Latina" because she cooks great Mexican food). | 40% |

Although the theoretical constructs around colonialism, post-colonialism, and oppression are important to me (and salient in my own worldview), they do reflect a particular historical and theoretical positionality that I do not think would be wise to tie to cultural competence. Certainly, people can be equally culturally responsive and yet adhere differently to these sociopolitical interpretations of history.

Experts noted the challenges inherent in measuring knowledge and one reviewer offered these ideas for additions to the knowledge scale:

knowledge [that] Whites suffer as a function of believing that they are superior and others inferior while they concomitantly depend of the cultural other for sustenance. Enslaved African women served as wet nurses, enslaved African men labored and thus both Enslaved African Men and Women built the economy that European Americans enjoy. European Americans are the real immigrants in the US and that land stolen does not constitute true ownership. Asians facilitated the building of the railroads that opened up travel from Eastern to the Western portion of the US. The Iroquois Nation had a bicameral system of government that serves as the basis for the current US system yet the indigenous people to this land are seen as inferior. There is a great psychological duress that European suffer as they believe the lie that they are superior. Questions about that would be interesting.

Finally, experts seemed to empathize with the difficulty in measuring specific skills and also advanced important ideas about items to add including consulting, participating in community events and other volunteerism, confronting racism, dominating conversations, and engaging in cultural appropriation, among others.

## Measures

In order to establish construct, concurrent, and discriminant validity, the ASK-G was measured along with ethnocultural empathy, colorblind racial attitudes, social dominance orientation, impressions of others, perceptions of discrimination, and a general index of personality. These constructs have often been used in conjunction with a conceptualization of cultural competence.

**Demographics.** In the survey that accompanied the ASK-G administration we asked participants to report on their age, gender identity, sexual orientation, employment, ethnicity, levels of education, employment, disability and health status, and family relationship status (parenting, partnered). These questions were selected for their inclusive structure [36].

**Colorblindness.** Colorblind attitudes were measured with the Color-Blind Racial Attitudes (CoBRAS) [32]. The CoBRAS is a 20 item self-report scale answered on a 6-point Likert-type scale that ranges from 1 (*strongly disagree*) to 6 (*strongly agree*). Responses are summed and possible values range from 20 to 120. Higher scores are indicative of higher endorsement of colorblind attitudes. The original scale underwent factor analysis, reliability checks, and thorough validity checks. Three factors emerged, racial privilege (score range: 7–42), institutional discrimination (score range: 7–42), and blatant racial issues (score range: 6–36). Concurrent, discriminant, criterion-related, and predictive validity were established for the measure [32]. Reliability estimates for the general scale and subscales in the current study were all adequate, $a$ = .71 - .84.

**Perceptions of discrimination.** Discrimination Perceptions is a subscale of the Multicultural Experiences Questionnaire (MEQ) [37]. The subscale is one item (item 16) that consists of a list 16 social groups (e.g., Native American, lesbians, right-wing groups) that participants are asked to rate on Likert-type scale based on the amount of discrimination participants believe each group faces from 1 (*no discrimination*) to 5 (*lots of discrimination)*. Reliability in the present study was good: $a$ = .93.

**Impression of social groups.** Social Group Impressions is a subscale of the MEQ [37]. This subscale is one item that uses a Likert-type scale to measure participants' feelings toward a list of 16 social groups (e.g., Black, women, fundamentalists) ranging from 1 (*very negative*) to 5 (*very positive*). Reliability in the present study was good: $a$ = .94.

**Ethnocultural empathy.** The Scale of Ethnocultural Empathy (SEE) is a 31-item scale rated from 1 (*strongly disagree that it describes me*) to 6 (*strongly agree that it describes me*) [33]. It has four subscales: empathic feeling and expression (15 items), empathic perspective taking (7 items), acceptance of cultural differences (5 items), and empathic awareness (4 items) in addition to returning a total scale score. The original scale development process revealed a stable factor structure, acceptable reliability estimates across subscales and the total score (range .73 - .91), and discriminant, concurrent, and criterion-related validity. The present study demonstrated adequate overall reliability ($a$ = .90) with subscales ranging from $a$ = .52 to .90.

**Social dominance orientation.** Social Dominance Orientation scale measures respondent's preference for inequality across ethnic/cultural groups across 14 items that are rated on a 1 (*very negative*) to 7 (*very positive*) Likert-type scale [34]. Higher scores indicate higher social dominance. The original scale validation showed strong reliability across 13 samples (*a*

range: .80 - .89) as well as evidence of discriminant and convergent validity [34]. Reliability for the current study was good, $a$ = .88.

**Personality.**　The Big Five Inventory (BFI) was used to measure personality [38]. This measure is a 44-item scale used to assess individuals on five dimensions of personality: extraversion (8 items), agreeableness (9 items), conscientiousness (9 items), neuroticism (8 items), and openness (10 items). Items are measured on a Likert-type scale from 1 (*disagree strongly*) to 5 (*agree strongly*). Reliability scores for the BFI have been found to range from .75 - .90 in US and Canadian samples with average around .80. Cronbach's alpha reliability scores for the present study BFI subscales ranged from .75 - .80. Scores are derived for the subscales by summing the corresponding items, with higher scores indicating more adherence to the personality trait being measured.

**Data analysis.**　All exploratory analyses for the current study were conducted in SPSS version 24 and confirmatory analyses were conducted in Mplus version 8.6. We evaluated results in two steps. We assessed confirmatory factor analysis (CFA) models to evaluate the hypothesized 3-factor structure of cultural competency using all items that were developed. We evaluated CFA models via model fit indices, and upon finding that the expected factor structure did not fit well (see results), we moved to an exploratory approach, first using parallel analysis to determine the number of factors underlying the items [39] then using exploratory factor analysis (EFA) to determine factor loading structure.

Parallel analysis uses a simulation-based approach to determine whether an extracted factor from the observed data has an eigenvalue statistically significantly greater than an extracted factor from the randomly simulated data. We used the rawpar.sps script from https://people. ok.ubc.ca/brioconn/nfactors/rawpar.sps to conduct this analysis. Further, a scree plot was assessed in conjunction with the parallel analysis to evaluate the number of unique factors above the "elbow" [40]. This method of extracting factors determines the point at which observed eigenvalues show a descending linear trend and additional factors add no meaningful variance to the model. An EFA was then conducted using results from the parallel analysis to inform the number of factors to be extracted. The EFA used principal axis factoring with varimax rotation. Items with rotated factor loadings of .50 or greater were retained in the final factor solution. We also evaluated results for promax rotation and report these findings in Appendix A.

## Results

We originally evaluated a 3-factor CFA, including all 81 items in analyses. The hypothesized three-factor scale that included all items fit the data poorly ($\chi^2$ = 7100.95, $df$ = 3399, $p$ < .001; RMSEA = .07; CFI = .60; SRMR = .09). Similarly, CFA models were evaluated separately for each of the 3 subscales. Subscale CFA models similarly resulted in poor model fit (awareness: $\chi^2$ = 1163.86, $df$ = 377, $p$ < .001; RMSEA = .10; CFI = .65; SRMR = .09; knowledge: $\chi^2$ = 856.34, $df$ = 275, $p$ < .001; RMSEA = .10; CFI = .71; SRMR = .08; skills: $\chi^2$ = 1127.17, $df$ = 405, $p$ < .001; RMSEA = .09; CFI = .74; SRMR = .08), suggesting that the items did not represent the hypothesized underlying latent dimensions as anticipated. Due to lack of model fit for the overall 3-factor CFA model, as well as lack of model fit for each of the subscale CFA models, we moved from a confirmatory approach to an exploratory approach to determine (a) if the items constructed more than a 3-factor solution, (b) which items loaded onto which factors, and (c) which items had the strongest loadings on each of the different factors. To reduce the number of items as well as determine the number of factors, we conducted parallel and exploratory factor analyses.

**Table 3. Parallel analysis results: Eigenvalues for raw data, random data means and 95th percentiles.**

| Factor | Raw data eigenvalue | Random data mean eigenvalue | Random data 95th percentile eigenvalue |
|---|---|---|---|
| 1 | 24.510 | 2.578 | 2.699 |
| 2 | 5.344 | 2.453 | 2.541 |
| 3 | 4.021 | 2.360 | 2.437 |
| 4 | 2.909 | 2.282 | 2.353 |
| 5 | 2.220 | 2.209 | 2.274 |
| 6 | 2.146 | 2.143 | 2.203 |
| 7 | 1.884 | 2.083 | 2.142 |
| 8 | 1.685 | 2.026 | 2.082 |
| 9 | 1.656 | 1.973 | 2.021 |
| 10 | 1.562 | 1.922 | 1.970 |

*Note*. Eigenvalues are only given for the first 10 factors and are not given for the remaining 74 potential factors. A four-factor solution best fits the data; all Raw Data Eigenvalues for the first four factors are greater than the Random Data 95th Percentile Eigenvalues.

## Parallel analysis

The parallel analysis utilized the same sample size ($N = 204$) and variable size ($k = 84$) as the initial dataset, used principal components analysis, specified 1,000 iterations, and evaluated results at the 95th percentile. Table 3 shows the eigenvalues of the observed data, the estimated mean eigenvalues of the random data, and the 95th percentile eigenvalues of the random data. When the eigenvalue from observed data is greater than the 95th percentile eigenvalue from the random data, the factor is statistically significant, and should be retained in future analyses. As shown in Table 3, results from the parallel analysis suggested a four-factor solution would best fit this data.

The scree plot also indicated that a four-factor solution fit the data (see Fig 1). We therefore used a four-factor solution in our exploratory factor analysis.

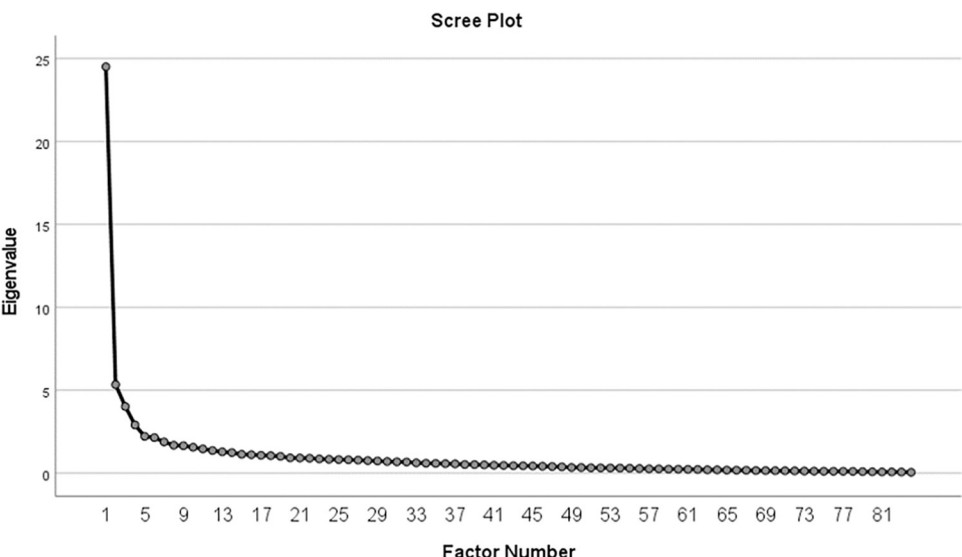

**Fig 1. Scree plot for ASK-G items.**

## Exploratory factor analysis

Before conducting the exploratory factor analysis (EFA), factorability of R was assessed, and results suggested sufficient levels of factorability using Kaiser-Meyer-Olkin (KMO) measure of sample adequacy, $KMO = .89$, and Bartlett's test of sphericity, $\chi^2$ ($df = 3486$) = 11,379, $p <$ .001. An EFA with principal axis factoring, varimax rotation, and four factors was then conducted. Items with loadings of at least .50 on any of the four factors were retained in the final factor structure. No items contained cross-loadings above .50. A total of 37 items were retained in the final factor structure. The pattern matrix of this solution is shown in Table 4 while the factor loadings for all items (including those items that were dropped) are shown in Appendix A. Results are also shown in Appendix A for the Promax rotated solution. We opted to keep the varimax rotated solution as the primary outcome reported in results since we wanted factors to be as distinct as possible (and since no large factor cross-loadings were obtained in the varimax solution).

The four-factor solution accounted for 44% of the variance. The residual R matrix was examined to evaluate the adequacy of the extraction procedure. Overall, the residual R matrix was composed of mostly zeroes ($M = .00$, *range*: -.21 to .26), but contained a moderate amount (33%) of residuals with values greater than .05. Factor 1 contained 13 items, Factor 2 contained 10 items, Factor 3 contained 7 items, and Factor 4 contained 7 items. Substantive meaning was applied to each factor based on the items within the factor and the correlations between the factors. Items within Factor 1 seemed to represent the predicted Knowledge subscale. Items from Factor 2 seemed to represent Awareness of Others. Items from Factor 3 represented Proactive Skills Development, and items in Factor 4 represented Awareness of Self.

Cronbach's alpha was computed for the overall scale and each of the subscales. The full scale reliability was very good, $a = .94$. Subscale reliabilities were also very good: Knowledge, $a = .92$; Awareness of Others, $a = .88$; Proactive Skills Development, $a = .87$; Awareness of Self, $a = .88$. Subscales were significantly correlated. Correlations between Knowledge and Awareness of Others was $r = .69$, $p < .001$, Knowledge and Proactive Skills Development, $r = .47$, $p <$ .001, Knowledge and Awareness of Self, $r = .39$, $p < .001$, Awareness of Others and Proactive Skills Development, $r = .28$, $p < .001$, Awareness of Others and Awareness of Self, $r = .37$, $p <$ .001, and Proactive Skills Development and Awareness of Self, $r = .43$, $p < .001$.

Average ASK-G results for the entire population are shown in Table 5. Correlations between the ASK-G and its subscales, BFI, CoBRAS, MEQ and SEE were conducted to determine the extent to which the ASK-G full and sub-scales were related to and different from related constructs (see Table 5). Correlations between the ASK-G and SEE, MEQ scales were mostly statistically significant, indicating concurrent validity. Correlations between the ASK-G and CoBRAS were all in the expected direction, but only half of the correlations were statistically significant, indicating some concurrent validity, but also some discriminant validity between these two scales. The ASK-G Awareness of Self subscale seemed to show different correlations with other variables than any of the other ASK-G subscales, and also had the lowest correlation with the ASK-G full scale, indicating that perhaps something unique was measured within this subscale.

## Discussion

We developed a general cultural competence measure based on the tripartite model from Sue's cultural competency model that can be used broadly across disciplines [4]. Many of the current empirically supported cultural competence measures are specific to practitioners or students and do not necessarily capture attitudes, knowledge, or skills relevant to interpersonal exchanges in the general population. The ASK-G shows promise as a tool for research into

**Table 4. Rotated ASK-G scale factor loadings for exploratory factor analysis with Kaiser-Varimax rotations.**

| Scale item | Factor | | | |
|---|---|---|---|---|
| | 1 | 2 | 3 | 4 |
| When I use an ethnic label to describe myself, I know what that label means to me. | **.633** | .210 | -.057 | .248 |
| I know about specific behaviors or routines that are specific to cultural groups other than my own (e.g., differences in how people greet each other). | **.646** | .226 | .328 | .151 |
| I know some history about people that belong to cultural groups different from my own. | **.615** | .305 | .086 | .136 |
| I know the difference between prejudice and discrimination. | **.541** | .207 | -.039 | .191 |
| I am familiar with religious beliefs and practices of cultural groups other than my own. | **.628** | .158 | .216 | .171 |
| I have learned about the history of a cultural group other than my own. | **.667** | .270 | .270 | .051 |
| I am familiar with important customs of a cultural group other than my own | **.623** | .177 | .280 | .154 |
| I can recognize the problem with applying stereotypes to specific cultural groups. | **.509** | .388 | .133 | .064 |
| I am able to take the perspective of a person from a culture other than my own. | **.582** | .399 | .154 | .070 |
| I am able to adjust my communication style when communicating with someone from a culture other than my own. | **.558** | .411 | .068 | .009 |
| I have attended ceremonies/celebrations (e.g., holiday celebrations, weddings, funerals, birthdays) from cultures different than my own. | **.569** | .250 | .349 | .017 |
| I have taken the time to learn about ways of being that are different from my own (e.g., religious traditions, coming-of-age ceremonies, medicinal approaches). | **.696** | .183 | .222 | .177 |
| There is no one "right" cultural perspective. | .178 | **.552** | .026 | .190 |
| There is no one "normal" culture. | .271 | **.562** | .066 | .077 |
| Racism affects everybody, not just underrepresented ethnic groups. | .222 | **.574** | -.161 | .074 |
| When I make a cultural misstep, I see that moment as a learning opportunity. | .256 | **.596** | .305 | .025 |
| There is room for me to grow in cultural competence. | .358 | **.610** | .155 | .033 |
| Some people have dietary restrictions specific to their cultural or religious upbringings. | .448 | **.512** | -.144 | .109 |
| Cultural competence is a lifelong journey rather than something with an end goal. | .334 | **.515** | .135 | .234 |
| When I say something that is offensive to another person, I can apologize even if I do not fully understand how I have offended them. | .272 | **.647** | .160 | .051 |
| I refrain from using certain words and phrases that I know may be offensive. | .381 | **.504** | .166 | -.025 |
| When I make a racist remark, I take time to reflect on the intention behind my comment and try to think of other ways I might get my point across. | .088 | **.538** | .334 | .066 |
| My cultural group membership has affected the opportunities that have been available to me. | .141 | .187 | **.580** | .348 |
| I listen to lectures or podcasts about cultural topics. | .228 | .005 | **.594** | .100 |
| I have joined a group that advocates for the rights of people in cultural groups different from my own. | .155 | .003 | **.740** | .057 |
| I openly speak a language other than my native language. | .178 | -.137 | **.654** | .141 |
| I regularly attend social action events (e.g., protests, town hall meetings) in my community. | .088 | -.105 | **.766** | .138 |
| I engage in advocacy work that advances the wellbeing of marginalized populations (e.g., homeless people, low income children). | .241 | .057 | **.644** | .109 |
| I confront racist comments in public settings made by strangers. | .354 | .172 | **.533** | -.091 |
| My cultural heritage has shaped who I am. | .194 | .136 | .058 | **.766** |
| My beliefs and values are rooted in my cultural background. | .199 | .262 | .143 | **.648** |
| My culture has an impact on the way I see the world. | .127 | .106 | .193 | **.730** |
| My culture has an impact on the way I think of others. | .010 | .064 | .275 | **.635** |
| My culture affects the way I behave toward others. | .054 | .079 | .459 | **.502** |
| My culture has shaped the way I see the world. | .223 | .146 | .163 | **.632** |
| My cultural values shape my assumptions about what is normal and abnormal. | .049 | .115 | .258 | **.612** |

**Table 5. Means, standard deviations of scales and correlation with ASK-G scales.**

|  | *M* | *SD* | ASK: AS | ASK: AO | ASK: PS | ASK: K | ASK-G |
|---|---|---|---|---|---|---|---|
| ASK-G |  |  |  |  |  |  |  |
| Awareness of self | 3.95 | 1.07 | 1.00** | .37** | .43** | .39** | .67** |
| Awareness of others | 4.49 | 0.89 | .37** | 1.00** | .28** | .69** | .78** |
| Skills, proactive | 3.21 | 1.22 | .43** | .28** | 1.00** | .47** | .69** |
| Knowledge | 4.28 | 0.94 | .39** | .69** | .47** | 1.00** | .88** |
| Full scale | 4.07 | 0.77 | .67** | .78** | .69** | .88** | 1.00** |
| Big 5 personality inventory |  |  |  |  |  |  |  |
| Extraversion | 24.31 | 5.54 | .01 | .09 | .27** | .21** | .20** |
| Agreeableness | 32.96 | 6.04 | -.02 | .36** | -.27** | .25** | .14 |
| Conscientiousness | 32.98 | 6.14 | .06 | .27** | -.20** | .27** | .16* |
| Neuroticism | 23.26 | 5.92 | -.08 | -.09 | .03 | -.09 | -.08 |
| Openness | 35.00 | 6.30 | .18** | .38** | .29** | .57** | .50** |
| Colorblind racial attitudes |  |  |  |  |  |  |  |
| Full scale | 68.91 | 16.03 | -.16* | -.29** | -.29** | -.25** | -.32** |
| Racial privilege | 25.69 | 7.81 | -.33** | -.24** | -.45** | -.24** | -.40** |
| Institutional discrimination | 25.76 | 7.61 | .12 | -.07 | -.15* | -.09 | -.07 |
| Blatant racism | 17.45 | 5.88 | -.14 | -.38** | .00 | -.24** | -.26** |
| Multicultural experiences Questionnaire |  |  |  |  |  |  |  |
| Discrimination perception | 47.64 | 14.38 | .11 | .27** | .14* | .24** | .26** |
| Social group impressions | 6.31 | 13.31 | .09 | .36** | .18* | .36** | .34** |
| Scale of ethnocultural empathy |  |  |  |  |  |  |  |
| Empathic feeling and expression | 3.99 | 0.96 | .24** | .66** | .35** | .62** | .64** |
| Empathic perspective taking | 3.84 | 0.74 | -.09 | .19** | .05 | .36** | .21** |
| Acceptance of cultural differences | 3.91 | 1.24 | -.24** | .09 | -.21** | .04 | -.08 |
| Empathic awareness | 4.00 | 1.22 | .37** | .55** | .33** | .54** | .60** |
| Full scale | 3.94 | 0.74 | .14* | .60** | .24** | .60** | .55** |

Note:

* $p < .05$

** $p < .01$; ASK AS = ASK-G Awareness of Self; ASK AO = ASK-G Awareness of Others; ASK PS = ASK-G Proactive Skills; ASK K = ASK-G Knowledge; ASK-G = ASK-G Full Scale.

cultural competence and evaluation of interventions. Our results further reflect that the Awareness of Self Subscale is measuring a unique aspect of cultural competence providing an even further exciting prospect for researchers, educators, and scholars to explore. It may also be a useful tool for self-evaluation [21].

The creation of the ASK-G was intentional and strategic in utilizing an expert panel of scholars across the US, experts of cultural competence within the research team, as well as outside perspectives from those passionate about cultural competence to provide a well-rounded set of scale items. The expert panel specifically provided a range of perspectives and expertise from various settings within psychology that allowed for items to be nuanced and targeted in the presentation of key cultural competence concepts. Research team members whose research specialties were outside of cultural competence were able to provide pragmatic feedback to ensure items were relevant and clear to those less, or unfamiliar with scholarly work in cultural competence. The feedback provided by the panel and research team members was able to be put into action by the cultural competence experts in the team so that items were conceptually sound, but also accessible to a general audience, which was key to this study.

Due to the variations of empirical descriptions of cultural competence the depth of analysis of experts was extremely helpful in developing items that mapped onto the tripartite model of cultural competence [4]. Consistent with prior measures using this conceptualization of cultural competency (e.g., Multicultural Awareness, Knowledge, and Skills) [26], we expected the ASK-G to be represented by three-factors (i.e., awareness, knowledge, skills). Contrary to our hypothesis, the three-factor solution did not fit the data well, which suggest a need for further exploration to identify the structure of the ASK-G. While prior measures have used this three-factor conceptualization among clinical professionals [26], other measures have opted for simpler two-factor structures (e.g., Multicultural Counseling Knowledge and Awareness Scale has two factors representing knowledge and awareness) or divided tripartite model components into multiple facets (e.g., Cultural Self-Efficacy Scale includes separate factors for knowledge of cultural concepts and knowledge of cultural patterns) [29, 41]. Thus, we were optimistic that an exploratory approach may identify a factor structure that provided better fit with multiple interpretable dimensions of cultural competency.

Exploratory factor analyses were conducted to determine the number of factors in the ASK-G scale and to identify items that may be eliminated to improve model fit and reduce respondent burden. We reduced the number of items from 84 to 37 and found a four-factor solution. The four-factor solution contained factors related to the intended structure: two Awareness factors (Awareness of Self and Awareness of Others), one Skills factor (Proactive Skills Development), and one Knowledge factor. All factors and the full scale had high reliability, which we expect was partly due to the strong theoretical approach used to create items. While prior measures have split facets of the tripartite model into multiple components, these factors were related to dimensions of knowledge (e.g., knowledge of cultural concepts and knowledge of cultural patterns) [41]. Prior measures have included a fourth factor alongside the tripartite model components representing multicultural relationships (i.e., Multicultural Counseling Inventory) [27], which most closely resembles the final factor solution for the ASK-G that includes two factors representing awareness of self and others. The ASK-G full scale and subscales showed strong concurrent validity with related constructs, namely ethnocultural empathy, discrimination perception, and social group impressions. Further, the scale showed adequate discriminant validity, as evidenced by weak, non-significant correlations with unrelated constructs, namely neuroticism and agreeableness. Overall, these results indicate that the ASK-G scale and subscales are both reliable and valid in measuring cultural competence.

A distinct strength of this study was the use of a sample with demographics representative of the U.S. population. This sample allowed the research team to gauge the usability of items in order to ensure the best fit statistically and practically. Researchers commonly use college student samples due to their accessibility and convenience [42]. However, a college sample would have been problematic during the current development due to the likelihood that college students have been introduced to diversity or cultural competence issues that may have skewed the results. This scale is also unique in targeting the general population making a college convenience sample inadequate for its development.

## Future directions

Now that we have developed a general scale, the true test of its utility will be achieved by way of predicting desired outcomes. In particular, this scale should be useful to administrators and educators seeking to verify the impact of training activities intended to improve cultural competence. In addition to time-limited workshops, many universities have semester-long courses on multicultural issues for which the ASK-G may be useful [21]. Evidence of shifts in cultural

competence would provide important data for the interventions and would also provide evidence of predictive validity for the ASK-G. In addition to these, cultural competence has been studied in very circumscribed contexts. An interesting avenue for future research would be to examine the ASK-G vis-à-vis the Implicit Associations Test. Because the ASK-G is intended for a general population, there may be relevant areas of study beyond intervention research, such as inter-group relationships in social and/or work settings, family cohesion, and/or friendship development. As the ASK-G focuses on race and ethnicity, future research might focus on adding items to include other dimensions of culture and identity (e.g., sexual orientation, gender identity). Finally, we evaluated the scale using a single, representative sample of the United States (i.e., exploratory factor analysis). The results from this study should be replicated in a confirmatory factor analysis for the general population and replicated across subpopulations.

## Limitations

The two parts of our study demonstrated the difficulty of assessing cultural competence skills, which was noted by the expert panel and emerged through the CFA and EFA analyses. The panel highlighted the difficulty in gauging the depth of participant responses in regard to items assessing exploration of other cultures through food and travel, which may be engaged thoughtfully or superficially. The EFA also demonstrated the difficulty of measuring skills with the emergence of two factors, proactive skills and awareness of others, from the items proposed by the research team to be the skills factor. The skills items represented within the scale are likely to capture "true positives" of those who engage the skills meaningfully, but may not capture other cultural competence skills that people may engage in.

## Conclusion

Overall, the ASK-G is a theoretically informed scale that was developed using a rigorous expert panel approach and was tested with a general population. The latter is of particular importance because the scale was developed for use with a general population. Rather than securing a sample of convenience (e.g., undergraduate students), we secured a national sample, representative of the general population on gender and ethnic lines. The resulting scale has strong psychometric properties and could be useful in evaluating diversity programming delivered to general audiences.

## Supporting information

**S1 Appendix. Factor loadings of original 81-item ASK-G scale.**
(DOCX)

## Author Contributions

**Conceptualization:** Melanie M. Domenech Rodríguez.

**Data curation:** Kaylee Litson.

**Funding acquisition:** Melanie M. Domenech Rodríguez.

**Investigation:** Melanie M. Domenech Rodríguez, Alexandra K. Reveles.

**Methodology:** Melanie M. Domenech Rodríguez, Alexandra K. Reveles, Kaylee Litson.

**Project administration:** Melanie M. Domenech Rodríguez.

**Resources:** Melanie M. Domenech Rodríguez.

**Software:** Kaylee Litson.

**Supervision:** Melanie M. Domenech Rodríguez.

**Visualization:** Kaylee Litson.

**Writing – original draft:** Melanie M. Domenech Rodríguez, Alexandra K. Reveles, Kaylee Litson, Christina A. Patterson, Alejandro L. Vázquez.

**Writing – review & editing:** Melanie M. Domenech Rodríguez, Alexandra K. Reveles, Kaylee Litson, Christina A. Patterson, Alejandro L. Vázquez.

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
