## [Decision Letter · Decision Letter 0]

31 May 2022

PONE-D-22-02341Development of the awareness, skills, knowledge: general (ASK-G) scale for measuring cultural competence in the general populationPLOS ONE

Dear Dr. Domenech Rodriguez,

Thank you for submitting your manuscript to PLOS ONE. After careful consideration, we feel that it has merit but does not fully meet PLOS ONE’s publication criteria as it currently stands. Therefore, we invite you to submit a revised version of the manuscript that addresses the points raised during the review process.

Your manuscript has been assessed by an expert reviewer, whose comments are appended below. The reviewer has highlighted concerns about several aspects of the methodology and theoretical basis of some decisions. Please ensure you respond to each point carefully in your response to reviewers document and modify your manuscript accordingly.

We look forward to receiving your revised manuscript.

Kind regards,

Joseph Donlan 

Editorial Office 

PLOS ONE

Journal Requirements:

This project was supported with research funds from the Culture & Mental Health Lab at Utah State University.

Reviewers' comments:

Reviewer's Responses to Questions

**Comments to the Author**

1. Is the manuscript technically sound, and do the data support the conclusions?

Reviewer #1: Partly

2. Has the statistical analysis been performed appropriately and rigorously? 

Reviewer #1: Yes

3. Have the authors made all data underlying the findings in their manuscript fully available?

Reviewer #1: Yes

4. Is the manuscript presented in an intelligible fashion and written in standard English?

Reviewer #1: Yes

5. Review Comments to the Author

Reviewer #1: I appreciate the opportunity to review this manuscript. This research addresses the measurement of cultural competence in the general population, a relevant and little explored topic. However, some aspects require clarification.

1) In the line 58, the authors argue "no know measures are designed for a general population". However, it is necessary to consider antecedents on measurements of intercultural personality and multicultural knowledge. The definition of these constructs and the items present in related measurements tools are similar to proposed in the manuscript reviewed. These aspects should be included in the manuscript.

2)In line 128, the authors mention a tool to measure CC in the general population. They should delve into it.

3) The section about the panel comments should be shortened as it is not a qualitative analysis of the information reported but only a selection of vignettes.

4) The argument for using Exploratory Factor Analysis (EFA) was the wide variety of conceptualization and definition of cultural competence. However, the items were developed using the Sue and Sue model, which has a precise definition of CC. Also, in line 155, they indicate, "we expect to find three factors". This hypothesis justify starting with a confirmatory model, and then (if it is necesary) an exploratory approach.

5) The authors estimated a sample to carry out EFA with 36 items, but the original questionnaire has more than 80. This aspect requires clarification since the sample size should consider the total number of items used in the first EFA. A sample of 200 people for an 80-item questionnaire might be insufficient.

6) The authors performed an EFA considering an orthogonal rotation, which assumes that the factors are independent of each other. This decision is not theoretically justified. They report moderate and high correlations between the factors, and the content of items does not suggest independent factors. An oblique rotation might be evaluated and the theoretical meaning of the items within the factors need to be revisited (there are very similar items in different factors, for example, "I able to adjust my communication style when communicating with someone from another culture…" (Knowledge) and "I refrain certain words and phrases that I know may be offensive" (Others Awareness))

7) The authors must include the first EFA with all items applied. This information could be in a supplementary file.

8) Given the nature of the items, some of them could load in two or more factors in a significant way (for example, loads >0.4). Report all factor loads.

9) The analysis section is integrated with the results section. I suggest adding an analysis section in the methodology for greater clarity.

10) I suggest incorporating some general estimation of the level of cultural competence (general and for each factor) achieved by the participants.

11) There is no theoretical definition of the factors in the final factor solution.

12) The discussion should place the findings in context by comparing them with other relevant literature. This aspect requires further development.

6. PLOS authors have the option to publish the peer review history of their article (what does this mean?). If published, this will include your full peer review and any attached files.

Reviewer #1: No

---

## [Author Response · Author response to Decision Letter 0]

29 Jul 2022

All comments are addressed in a detailed letter attached to the submission.

---

## [Decision Letter · Decision Letter 1]

30 Aug 2022

Development of the awareness, skills, knowledge: general (ASK-G) scale for measuring cultural competence in the general population

PONE-D-22-02341R1

Dear Dr.Rodriguez

We’re pleased to inform you that your manuscript has been judged scientifically suitable for publication and will be formally accepted for publication once it meets all outstanding technical requirements.

Kind regards,

Ali Montazeri

Academic Editor

PLOS ONE

Additional Editor Comments (optional):

Reviewers' comments:

Reviewer's Responses to Questions

**Comments to the Author**

1. If the authors have adequately addressed your comments raised in a previous round of review and you feel that this manuscript is now acceptable for publication, you may indicate that here to bypass the “Comments to the Author” section, enter your conflict of interest statement in the “Confidential to Editor” section, and submit your "Accept" recommendation.

Reviewer #1: All comments have been addressed

2. Is the manuscript technically sound, and do the data support the conclusions?

Reviewer #1: Yes

3. Has the statistical analysis been performed appropriately and rigorously? 

Reviewer #1: Yes

4. Have the authors made all data underlying the findings in their manuscript fully available?

Reviewer #1: Yes

5. Is the manuscript presented in an intelligible fashion and written in standard English?

Reviewer #1: Yes

6. Review Comments to the Author

Reviewer #1: The authors considered all comments. The manuscript has improved its overall quality. I thank the appendix information and the clarity of the results.

7. PLOS authors have the option to publish the peer review history of their article (what does this mean?). If published, this will include your full peer review and any attached files.

Reviewer #1: No

---

## [Editor Report · Acceptance letter]

1 Sep 2022

PONE-D-22-02341R1 

Development of the awareness, skills, knowledge: general (ASK-G) scale for measuring cultural competence in the general population 

Dear Dr. Domenech Rodríguez:

I'm pleased to inform you that your manuscript has been deemed suitable for publication in PLOS ONE. Congratulations! Your manuscript is now with our production department. 

Kind regards, 

on behalf of

Professor Ali Montazeri 

Academic Editor

PLOS ONE